# CyberV: A Cybernetic Framework for Enhancing Logical Reasoning in Video Understanding

## Abstract

Current Multimodal Large Language Models (MLLMs) may struggle with tasks requiring deep logical reasoning about video content, primarily stemming from the feed-forward processing nature, which limits their ability for self-correction and iterative refinement. To address these limitations, we propose a novel framework inspired by cybernetic principles, redesigning video MLLMs as adaptive systems capable of self-monitoring, self-correction, and dynamic resource allocation during inference. Our approach, **CyberV**, introduces a cybernetic loop consisting of an MLLM Inference System, a Sensor, and a Controller. Specifically, the sensor monitors MLLM forward processes. It collects intermediate interpretations, such as attention drift, then the controller determines when and how to trigger self-correction and generate feedback to guide the next round. This test-time adaptive scaling framework enhances frozen MLLMs without requiring training or additional components. Experiments demonstrate significant improvements on complex reasoning benchmarks: CyberV boosts Qwen2.5-VL-7B by **8.3%** and InternVL3-8B by **5.5%** on VideoMMMU, surpassing the competitive proprietary model GPT-4o. When applied to Qwen2.5-VL-72B, it yields a **10.0%** improvement, achieving performance even comparable to human experts. Furthermore, on other reasoning-focused benchmarks, our method shows consistent gains of 4.6% on the multiple-choice question section of MMVU and 2.4% on MMR-V, highlighting its robustness in enhancing logical reasoning for video understanding. The code will be released to support further research.

## 1 Introduction

Understanding dynamic visual scenes in videos is a fundamental challenge, with applications ranging from autonomous driving to content analysis and human-robot interaction. Multimodal Large Language Models have recently emerged as a powerful paradigm, demonstrating impressive capabilities by integrating pre-trained large language models with visual encoders to process and reason about video content Wang et al. (2023a); Bai et al. (2023); Team et al. (2024); Zhang et al. (2024b); Fei et al. (2025). However, deploying these models effectively, particularly for tasks that demand deep, multi-step logical reasoning on video content, presents significant hurdles. Current MLLMs often struggle with the computational demands of processing extended video streams (test-time scaling), exhibit brittleness to variations or unexpected events in the input (lack of robustness), and are prone to generating inaccurate, inconsistent, or hallucinatory interpretations (limited accuracy) Zhang et al. (2024b); Li et al. (2024a); Fei et al. (2024); Hu et al. (2025a); Han et al. (2024). In particular, directly applying reflection prompts performs poorly in video understanding (Figure 1). Even state-of-the-art MLLMs struggle to scale their capabilities effectively. Moreover, simply using chain-of-thought prompting may also degrade the model's perceptual abilities (Appendix A.2).

We argue that these limitations stem from the feed-forward processing pipeline inherent in current MLLM architectures. These models typically process videos in a single, often computationally intensive, pass, lacking dynamic adaptation, self-correction, or targeted analysis based on evolving understanding or specific task demands. This contrasts sharply with biological systems, which continuously use feedback to regulate behavior and adapt to complex environments.

To address this gap, we propose incorporating principles from cybernetics – the study of control, communication, and self-regulation in systems Wiener (1948); Ashby (1956) – into the design of

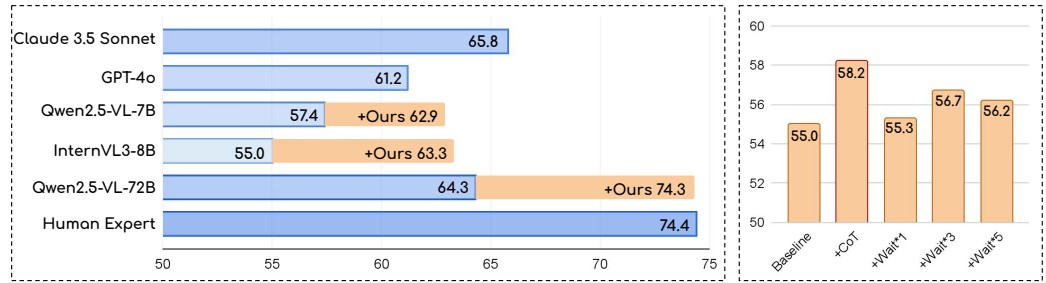

Figure 1: Performance on the VideoMMMU benchmark. **Left:** CyberV boosts small open-source models with only 7B parameters to outperform GPT-4o; with a larger model, CyberV surpasses the prior state-of-the-art and approaches the human level. **Right:** CoT reasoning improves results when using Qwen2.5-VL-7B, but multi-round reflection via "Wait" degrades performance.

MLLMs for video understanding. Cybernetics provides a rich theoretical framework centered on feedback loops, adaptive control, and goal-oriented behavior, enabling stable and effective operation in complex, dynamic settings. We hypothesize that by redesigning video MLLMs as cybernetic systems, capable of self-monitoring, self-correction, and adaptive resource allocation during inference, we can significantly enhance their performance.

Implementing these principles, we introduce **CyberV**, a framework structured as a dynamic feedback loop to create more robust and accurate MLLMs for video reasoning. The core of this cybernetic system consists of three components: the MLLM Inference System, the Sensor, and the Controller. Specifically, we generate responses by applying various scaling strategies within the MLLM inference system. These responses may come directly from the base MLLM model or be enhanced through techniques such as chain-of-thought prompting or the incorporation of key frames. The sensor monitors the inference processes and collects intermediate signals, such as the attention drift among different outputs and the prediction results, as evidence for the re-analysis of the controller. Given the evidence, the controller calculates the confidence score and determines whether to terminate the loop or to forcibly trigger a self-correction process to avoid unreliable responses during inference. If the termination condition is not met, the controller takes action by injecting the generated feedback into the MLLM's input for the next round of inference, closing the entire cybernetic loop.

We demonstrate the efficacy of the proposed cybernetic mechanisms on challenging video understanding benchmarks that require complex reasoning. Our experiments demonstrate that CyberV can remarkably improve the performance of a relatively small model by a large margin. As shown in Figure 1, it improves the accuracy of Qwen2.5-VL-7B by 8.3% and InternVL3-8B by 5.5%, allowing both models to surpass GPT-4o on the VideoMMMU benchmark Hu et al. (2025b). Furthermore, when applied to a larger model, Qwen2.5-VL-72B, our approach yields a 10.0% improvement over the baseline, achieving performance comparable to human experts. To further validate its effectiveness, our method also delivers consistent improvements on other video reasoning benchmarks. For instance, based on the Qwen2.5-VL-7B model, it achieves gains of 4.6% on the multiple-choice question section of MMVU Zhao et al. (2025) and 2.4% on MMR-V Zhu et al. (2025b). Extensive ablation studies show the effectiveness of each component in our CyberV framework. Our **contributions** can be summarized as follows:

- We propose **CyberV**, a test-time adaptive scaling framework based on cybernetic feedback control that enhances the reasoning abilities of frozen MLLMs without training or extra components (vision expert models).

- We introduce an **attention-based** monitoring mechanism and an adaptive scoring controller that jointly govern strategy selection during inference.

- CyberV empowers small models to outperform proprietary systems like GPT-4o, and enables large open-source models to achieve state-of-the-art results on VideoMMMU. Extensive experiments demonstrate the effectiveness and robustness of our approach on **complex video reasoning tasks**.

## 2 RELATED WORK

**Multi-modal Large Language Models in Video.** Multi-modal large language models (MLLMs) have seen growing attention in the video domain, with numerous video-specific models Zhang et al. (2024b; 2025a); Li et al. (2024d); Shu et al. (2024); Zhang et al. (2024a); Wang et al. (2025) and foundation models like the Qwen-VL Bai et al. (2023); Wang et al. (2024); Bai et al. (2025) and InternVL Chen et al. (2024c;a); Zhu et al. (2025a) series being developed. While these architectures demonstrate impressive perception capabilities, most struggle with complex reasoning over video content. To address this gap, recent efforts Guo et al. (2025a); Muennighoff et al. (2025); Shao et al. (2024) introduce reinforcement learning and test-time scaling strategies to enhance language model reasoning, and have been extended to video understanding through models such as Video-R1 Feng et al. (2025), VideoChat-R1 Li et al. (2025), and TinyLLaVA-Video-R1 Zhang et al. (2025b). In parallel, video chain-of-thought (CoT) prompting methods such as Video-of-Thought Fei et al. (2024), Chain-of-Shot Hu et al. (2025a), Logic-in-Frames Guo et al. (2025b) decompose complex video reasoning tasks into manageable sub-problems, addressing them step-by-step from low-level perception to high-level cognition. However, most existing approaches require supervised post-training or auxiliary models. In contrast, our work explores a training-free, single-model strategy that performs strongly on logic-based video tasks, suggesting that simple, modular inference techniques can yield robust multimodal reasoning.

**Test Time Scaling.** This direction Snell et al. (2024); Muennighoff et al. (2025); Liu et al. (2025) is a promising strategy for improving LLM performance by allocating more compute during inference. TTS methods generally fall into two categories: **Sequential Scaling**, which prolongs the reasoning process (e.g. chain of thought Wei et al. (2022), reflection Muennighoff et al. (2025)); and **Parallel Scaling**, which explores multiple reasoning paths and selects the best (e.g. Best-of-N Brown et al. (2024)). Parallel methods are often combined with sequential strategies to form complex search procedures, such as tree search Liu et al. (2025), with majority voting Wang et al. (2023b), output reward models (ORMs) Xin et al. (2024), and process reward models (PRMs) Uesato et al. (2022), often used to verify reasoning steps. While effective in textual tasks Muennighoff et al. (2025); Liu et al. (2025), TTS remains underexplored in video understanding. Our findings suggest that directly applying existing techniques often yields limited gains, highlighting the need for modality-aware scaling strategies.

**Cybernetics in Machine Learning and AI Systems.** Cybernetics, first formalized by Wiener Wiener (1948), provides a theoretical framework for self-regulating systems composed of three core components: a sensor for observing system states, a controller for decision-making, and a plant or system being controlled McCulloch & Pitts (1943). While influential in early AI research Ashby (1956); von Foerster (1952), its integration into modern deep learning remains limited. Some recent works have introduced feedback mechanisms into neural architectures Huang et al. (2020); Zhang & Lu (2023), but these often require architectural modifications or specialized training. In contrast, our approach utilizes the MLLM Inference System, Sensor, and Controller to apply cybernetic principles, significantly improving performance without the need for additional training.

## 3 METHOD

### 3.1 CYBERNETIC VIEW FOR VIDEO TEST-TIME SCALING

Test-time scaling for multimodal large language models (MLLMs), particularly in logic-based video understanding tasks, presents significant challenges. Unlike text-only reasoning, where techniques such as chain-of-thought and self-reflection prompting can be applied with moderate success, MLLMs face greater complexity due to the temporal, visual, and semantic richness of video data. Existing approaches Bai et al. (2025); Zhu et al. (2025a); Li et al. (2024a) apply static inference strategies that do not adapt to input difficulty, uncertainty, or reasoning failure, leading to inefficiencies and suboptimal performance. To overcome these limitations, we propose a cybernetic framework that transforms test-time inference into a feedback-driven, adaptive process inspired by the principles of control and regulation in cybernetics. We conceptualize video reasoning during inference as a closed-loop control system consisting of three interdependent components:

**MLLM Inference System:** This is the plant in the cybernetic loop, responsible for executing inference over multimodal input. Responses serve as raw material for further evaluation.

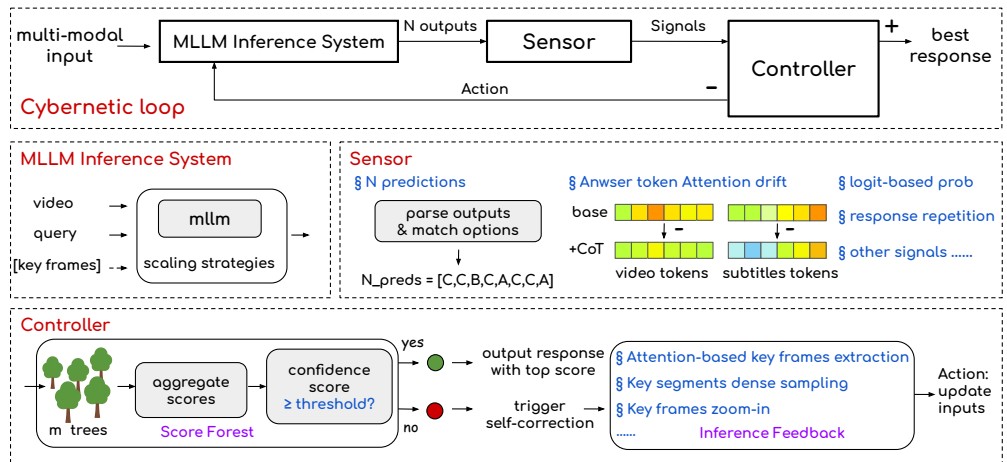

Figure 2: **Overall framework of CyberV.** CyberV models test-time video understanding as a closed-loop cybernetic process with three modules: ***MLLM Inference System***, ***Sensor***, **and** ***Controller***. The inference system executes inference scaling strategies with a frozen MLLM, generating N outputs. The sensor monitors the MLLM forward process, extracting signals such as parsed prediction and attention drift. The controller uses a **Score Forest** to evaluate response reliability, triggering self-correction via the **Inference Feedback** module if confidence is low. The updated input is then used to re-invoke the MLLM. This feedback loop enables adaptive and robust test-time reasoning.

**Sensor:** The Sensor monitors the Inference System and extracts key signals such as predicted options and attention drift among different responses. These signals reflect the inference reliability, forming the basis for later decision-making.

**Controller:** The Controller is the central decision-making unit of the cybernetic system. It receives multiple signals from the Sensor, and evaluates the confidence of each candidate response using a rule-based scoring ensemble. Based on a thresholding policy, it decides whether to accept the output or trigger further inference with generated feedback.

Specifically, given a video $V$, a query $q$ that includes the question and the video subtitles (if available), and a strategy set $\Pi = \{\pi_1, \ldots, \pi_N\}$, the frozen model $\mathcal{M}$ generates candidate responses $\{r_i\}_{i=1}^N$, where each $r_i$ can be expressed as $r_i = \mathcal{M}_{\pi_i}(q, V)$. These responses are then passed back to the Sensor and Controller for evaluation, forming a closed feedback loop. This iterative process allows the model to monitor its interpretations, adjust inference paths in real time, and allocate computational resources more efficiently based on task relevance and video complexity. Unlike prior test time scaling approaches Muennighoff et al. (2025) that apply fixed reasoning templates regardless of context, our method dynamically modulates processing depth and focus, enhancing both accuracy and robustness.

As illustrated in Figure 2, the proposed framework **CyberV** implements this **cybernetic loop** to adaptively scale inference at test time without any parameter updates or supervision. It empowers a frozen model to handle diverse video understanding tasks by actively managing its reasoning process in response to control signals. The pseudo-code of CyberV is presented in Appendix A.3.

## 3.2 MLLM INFERENCE SYSTEM: EXECUTING TEST-TIME SCALING STRATEGIES

The MLLM inference system, the plant in our cybernetic loop, executes diverse test-time scaling strategies on multimodal input. We adopt the Best-of-N (BoN) scheme that executes N forward passes in parallel to generate a set of candidate responses. Each inference path may vary in its configuration, including direct answer using the base model, chain-of-thought prompting to encourage reasoning, such as "Thinking Process:", or the incorporation of visually enhanced inputs such as injected key frames. This design enables the system to adaptively combine structured reasoning and perceptual reinforcement based on task uncertainty. Compared to more complex search strategies, such as performing tree search, Best-of-N offers a simpler yet effective alternative.

### 3.3 SENSOR: SIGNAL EXTRACTION FROM MLLM FORWARD PROCESSES

The Sensor monitors the forward execution of the MLLM and extracts informative signals that serve as the basis for confidence evaluation and feedback decisions.

One key signal is the predicted answer label $\{\hat{y}_n\}_{n=1}^{N}$, obtained by parsing $N$ textual responses $\{r_n\}_{n=1}^{N}$, where $\hat{y}_n \in \mathcal{C}$, and $\mathcal{C}$ is the candidate set of choices, e.g. A, B, C, D etc. The parsing relies on explicit pattern matching.

Additionally, the Sensor evaluates the model's perceptual behavior by quantifying attention drift. As the video and subtitles can be segmented according to the number of frames and timestamps, we compare the attention distribution over these segments across two settings: a base response and a chain-of-thought prompting variant. Specifically, the video is divided into $K_1$ segments and the subtitles into $K_2$ segments. For each attention head $h \in \{1, \ldots, H\}$, where $H$ is the total number of attention heads, we define the attention scores from the answer token to the video and subtitle segments in the final layer as $\mathbf{A}_h^{\text{video}} \in [0,1]^{1 \times K_1}$ and $\mathbf{A}_h^{\text{sub}} \in [0,1]^{1 \times K_2}$. The attention drift signal $\Delta^{\text{video}} \in [-1,1]^{1 \times K_1}$ for video part and $\Delta^{\text{sub}} \in [-1,1]^{1 \times K_2}$ for subtitles part is defined as:

$$\Delta^{\text{video}} = \frac{1}{H} \sum_{h=1}^{H} \left( \mathbf{A}_{h,\text{cot}}^{\text{video}} - \mathbf{A}_{h,\text{base}}^{\text{video}} \right), \quad \Delta^{\text{sub}} = \frac{1}{H} \sum_{h=1}^{H} \left( \mathbf{A}_{h,\text{cot}}^{\text{sub}} - \mathbf{A}_{h,\text{base}}^{\text{sub}} \right). \tag{1}$$

where the subscript "base" and "cot" refer to the base and chain-of-thought model responses, respectively. The subtitle part is not considered when the video has no audio. We use the "base" response as an anchor because its attention often reflects a more direct, foundational grounding on visual evidence. Our goal is to diagnose whether the complex reasoning process induced by "cot" prompting inadvertently causes the model's focus to drift away from this crucial initial grounding. Therefore, a large negative value of $\Delta$ on a certain segment indicates that the CoT process has significantly distracted the model's attention, suggesting degraded perceptual grounding. This mechanism is designed to identify cases where the base model can answer correctly, but is misguided by the CoT-induced reasoning path. The controller then uses this signal to trigger a self-correction via the feedback loop.

Beyond attention and answer prediction, the Sensor can also collect other forward-pass signals to characterize response quality. For example, the softmax confidence of the predicted token, logit stability across strategies, and repetition patterns in the response text. Together, these features offer a rich diagnostic view of the model's current inference behavior and serve as input to the control mechanism that governs adaptive reasoning.

### 3.4 CONTROLLER: DECISION MAKING AND FEEDBACK CONSTRUCTION

The Controller governs the adaptive reasoning process by making two key decisions: whether the current inference results are sufficiently reliable for output, and if not, how to generate actionable feedback to revise the model's input for the next iteration. It comprises two coordinated modules: a **Score Forest** for response evaluation and an **Inference Feedback** module for corrective input construction.

**Score Forest: Confidence-Aware Evaluation and Thresholding.** Given $N$ candidate responses and multiple signals from the sensor, the Score Forest assigns each response a multi-dimensional score vector $\mathbf{s}_n = (s_{n,1}, \ldots, s_{n,m}) \in [0,1]^m$, capturing semantic, probabilistic, and attention-related qualities. Here, $m$ is the number of trees in the forest, where each tree maps extracted signals to a score via a distinct mechanism. In our implementation ($m = 5$), the rule-based scoring mechanisms include: an **Attention Retention Score** from the attention drift signal $\Delta$; the raw **Answer Confidence Score** based on softmax probability of the predicted answer token.; a binary **Confidence Stability Score** to penalize significant drops after CoT; a **Relative Rank Score** formulated as $1 - (\text{rank}_n - 1)/N$ derived from confidence order; and a binary **Text Repetition Score** to penalize redundancy where a long sequence is repeated beyond five times. The final score $S_n$ for each response is computed as the average of these individual scores: $S_n = \frac{1}{m} \sum_{i=1}^{m} s_{n,i}$.

Furthermore, we can calculate the top-scoring option based on these scores. The score of the best option, $TopScore \in [0, N]$, can be calculated by $TopScore = \max_{c \in \mathcal{C}} \sum_{n:\hat{y}_n=c} S_n$. If the top score satisfies the confidence threshold, i.e., $TopScore \geq \tau \cdot N$, where the threshold $\tau \in [0, 1]$,

Table 1: **Performance on the VideoMMMU benchmark (accuracy %).** The results are grouped by evaluation track (Perception, Comprehension, Adaptation) and academic discipline (Art, Business, Science, Medicine, Humanities, and Engineering). Yellow rows indicate open-source MLLMs, while blue rows indicate proprietary models. "*w/sub*" indicate our baselines with subtitle input.

| Model | Overall | Results by Track | | | Results by Discipline | | | | | |
|---|---|---|---|---|---|---|---|---|---|---|
| | | Percep. | Compr. | Adapt. | Art. | Biz. | Sci. | Med. | Hum. | Eng. |
| Human Expert | 74.4 | 84.3 | 78.7 | 60.3 | 81.0 | 78.8 | 74.2 | 70.5 | 84.8 | 69.9 |
| LLaVA-OneVision-7B Li et al. (2024a) | 33.9 | 40.0 | 31.0 | 30.7 | 49.2 | 29.6 | 34.9 | 31.8 | 46.7 | 29.2 |
| VILA1.5-40B Lin et al. (2024) | 34.0 | 38.7 | 30.7 | 32.7 | 57.1 | 27.3 | 23.5 | 38.0 | 41.9 | 32.5 |
| LLaVA-Video-7B Zhang et al. (2024b) | 36.1 | 41.7 | 33.3 | 33.3 | 65.1 | 34.1 | 32.6 | 42.6 | 45.7 | 27.4 |
| InternVL2-8B Chen et al. (2024b) | 37.4 | 47.3 | 33.3 | 31.7 | 55.6 | 34.1 | 30.3 | 34.1 | 41.9 | 38.1 |
| LLaVA-OneVision-72B Li et al. (2024a) | 48.3 | 59.7 | 42.3 | 43.0 | 61.9 | 46.2 | 40.2 | 54.3 | 60.0 | 44.0 |
| LLaVA-Video-72B Zhang et al. (2024b) | 49.7 | 59.7 | 46.0 | 43.3 | 69.8 | 44.7 | 41.7 | 58.9 | 57.1 | 45.1 |
| Gemini 1.5 Flash Team et al. (2024) | 49.8 | 57.3 | 49.0 | 43.0 | 63.5 | 53.0 | 43.2 | 49.6 | 59.1 | 45.7 |
| Aria Li et al. (2024b) | 50.8 | 65.7 | 46.7 | 40.0 | 71.4 | 47.7 | 44.7 | 58.9 | 62.9 | 43.7 |
| Gemini 1.5 Pro Team et al. (2024) | 53.9 | 59.0 | 53.3 | 49.3 | 57.1 | 59.1 | 49.1 | 57.4 | 58.1 | 50.3 |
| Qwen2.5-VL-7B Bai et al. (2025) (w/ sub) | 55.0 | 72.7 | 53.7 | 38.7 | 73.0 | 56.1 | 46.2 | 58.1 | 73.3 | 47.8 |
| InternVL3-8B Zhu et al. (2025a) (w/ sub) | 57.4 | 77.0 | 49.7 | 45.7 | 61.9 | 59.1 | 53.0 | 60.5 | 74.3 | 51.3 |
| GPT-4o OpenAI (2024) | 61.2 | 66.0 | 62.0 | 55.7 | 69.5 | 66.9 | 51.6 | 64.8 | 69.5 | 57.1 |
| Qwen2.5-VL-72B Bai et al. (2025) (w/ sub) | 64.3 | 84.7 | 63.0 | 45.3 | 79.4 | 66.7 | 62.9 | 68.2 | 81.9 | 54.3 |
| Claude 3.5 Sonnet Anthropic (2024) | 65.8 | 72.0 | 69.7 | 55.7 | 66.7 | 75.0 | 56.1 | 58.1 | 75.2 | 66.1 |
| InternVL3-8B Zhu et al. (2025a) (+Ours) | 62.9 (+5.5) | 77.3 | 60.3 | 51.0 | 65.1 | 67.4 | 62.1 | 62.0 | 80.0 | 56.0 |
| Qwen2.5-VL-7B Bai et al. (2025) (+Ours) | 63.3 (+8.3) | 78.0 | 62.0 | 50.0 | 76.2 | 65.9 | 54.5 | 64.3 | 75.2 | 59.3 |
| Qwen2.5-VL-72B Bai et al. (2025) (+Ours) | **74.3 (+10.0)** | **85.7** | **76.3** | **61.0** | **82.5** | **78.0** | **68.2** | **78.3** | **83.8** | **69.3** |

the controller selects the top-score answer as the final output. Otherwise, the low confidence score indicates unreliable initial reasoning, and the controller triggers the Inference Feedback module to initiate a corrective update. Note that the widely used majority voting policy can be viewed as a special case of the Score Forest, where the score $S_n = 1$ for each response $n$ and the threshold $\tau = 0$.

**Inference Feedback: Visual Correction for Self-Revision.** When confidence is insufficient, the Controller invokes the Inference Feedback module to construct enhanced input that guides the next round of reasoning. This module identifies the top-k visual and subtitle segments that exhibit the greatest decrease in attention. Specifically, we define: $\mathcal{I}_{\text{video}} \subseteq \{1, \ldots, K_1\}$ and $\mathcal{I}_{\text{sub}} \subseteq \{1, \ldots, K_2\}$, where each set contains the indices of the top-k segments with the largest attention decrease:

$$\mathcal{I}_{\text{video}} = \text{TopK-Indices}(-\Delta_j^{\text{video}}), \quad \mathcal{I}_{\text{sub}} = \text{TopK-Indices}(-\Delta_j^{\text{sub}}). \tag{2}$$

For $\mathcal{I}_{\text{video}}$, we can directly extract the corresponding frames through the indices. For $\mathcal{I}_{\text{sub}}$, we trace their timestamps to locate the aligned frames. The union of these yields the final set of key frames. To restore the model's degraded attention due to reasoning steps, the identified key frames can be seamlessly re-integrated into the original input sequence.

To further refine the model's focus, we also support more visual content enhancement methods. Temporally, we perform dense sampling around selected key frames while sparsely sampling elsewhere. Spatially, we apply a zoom-in to emphasize evidence-rich regions by computing region-question relevance. To achieve this, key frames are partitioned into a grid of regions at multiple granularities. We then calculate the CLIP-based similarity between the question and each region, and the region with the highest relevance is cropped and used as an enhanced visual input. These enhanced inputs are then sent back to the MLLM inference system, enabling the model to refocus on critical evidence.

## 4 EXPERIMENTS

**Benchmarks.** To rigorously evaluate our framework's ability to enhance logical reasoning in video understanding tasks, we conduct experiments on three related benchmarks. **VideoMMMU** Hu et al. (2025b) consists of 300 expert-level educational videos and 900 questions spanning six academic disciplines, making it the ideal testbed for an in-depth evaluation of our framework's performance on knowledge-intensive reasoning. **MMVU** Zhao et al. (2025) tests expert-level reasoning with 1,000 questions (validation set) on professional videos from 27 sub-disciplines, requiring the application of deep domain knowledge. **MMR-V** Zhu et al. (2025b) measures deep deductive reasoning with 1,257 multiple-choice questions on 317 diverse videos. It challenges models with long-range, multi-hop

Table 2: Performance on other video reasoning benchmarks (accuracy %). "Val" means validation set, "MCQ" means multi-choice, and "w/sub" means adding subtitles in the prompt. The base model is Qwen2.5-VL-7B.

| Model | MMVU (Val, MCQ) | MMR-V (w/sub) | | |
|---|---|---|---|---|
| | Overall | Implicit | Explicit | Overall |
| Base | 62.1 | 42.4 | 26.4 | 38.3 |
| +Ours | **66.7 (+4.6)** | **44.0** | **30.9** | **40.7 (+2.4)** |

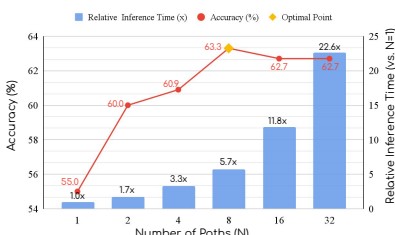

Figure 3: **Analysis of Performance vs. Efficiency.** The optimal balance is achieved at N=8, which is highlighted.

logic to infer unstated information against confusing distractors. By evaluating on these benchmarks, we demonstrate our framework's ability to enhance a spectrum of logical reasoning skills, from applying domain knowledge to performing deep deductions.

**Implementation Details.** We adopt Qwen2.5-VL Bai et al. (2025) and InternVL3 Zhu et al. (2025a) as the base models for all evaluations, using 64 uniformly sampled frames per video for Qwen2.5-VL and 32 frames for InternVL3. Subtitles, when needed, are extracted using Faster-Whisper Large-v3.[1] We adopt a two-round Best-of-N scheme across all benchmarks. In the Controller, response confidence is estimated via the Score Forest, and direct key frame injection is employed as the feedback mechanism. For all three benchmarks, we set $N = 8$ and $\tau = 0.3$ in the first round, using one base strategy and seven chain-of-thought (CoT) variants. In the second round, we set $N = 1$ and $\tau = 0$. We use accuracy as the evaluation metric for all benchmarks. All experiments are conducted on 8 GPUs, each equipped with 80 GB of memory.

## 4.1 MAIN RESULTS

Table 1 shows the primary results of our method on the **VideoMMMU** Hu et al. (2025b) benchmark. We compare our approach against two major categories of models: (1) open-source MLLMs, and (2) proprietary models such as GPT-4o and Claude 3.5 Sonnet. Human expert performance is also provided for reference. Our results show that **CyberV consistently enhances model performance** across a range of model scales. When applied to Qwen2.5-VL-7B, it achieves a notable +8.3% improvement over the base model, surpassing GPT-4o by 2.1% and approaching Claude 3.5 Sonnet. On InternVL3-8B, it brings a +5.5% gain, outperforming GPT-4o by 1.7%. For Qwen2.5-VL-72B, CyberV further boosts performance by +10.0%, exceeding Claude 3.5 Sonnet by 8.5% and reaching accuracy on par with human experts. These results show that even smaller open-source models can outperform proprietary LLMs through effective cybernetic inference-time scaling. In addition to the overall accuracy, we observe that CyberV is especially effective on the **comprehension** and **application** tracks, where reasoning and knowledge transfer are essential. By discipline, the most significant gains occur in **business**, **science**, **medicine**, and **engineering**, indicating that our method is crucial for knowledge-intensive video reasoning tasks.

To further demonstrate the framework's effectiveness, we present additional results on two other video reasoning benchmarks, MMVU Zhao et al. (2025) and MMR-V, as shown in Table 2. For these experiments, we use Qwen2.5-VL-7B as the base model. On **MMVU**, our method achieves a significant **+4.6%** improvement. Note that here we focus specifically on its multiple-choice question (MCQ) section, as our current rule-based Score Forest is designed to score responses from a discrete set of candidate answers, making it less suited for free-form questions. On **MMR-V**, CyberV delivers a solid **+2.4%** overall gain, with consistent improvements observed across both its explicit and implicit reasoning sub-tasks. These results confirm that our framework provides stable and effective performance enhancements across a variety of video reasoning challenges. Furthermore, the results and discussion of perception-heavy video understanding benchmarks are provided in Appendix A.2.

---

[1]https://github.com/SYSTRAN/faster-whisper

Table 3: **Ablation Study on MLLM Inference System and Sensor.** Accuracy (%) on VideoMMMU under different inference configurations and search schemes for MLLM inference system, and different sources of attention drift for Sensor.

(a) Impact of different scaling strategies.

| Strategy | Acc (%) |
|---|---|
| Base | 48.6 |
| +Subtitles | 55.0 (+6.4) |
| +CoT | 58.2 (+9.6) |
| +Key Frames | 60.0 (+11.4) |

(b) Comparison of more search schemes. "KF" means key frames.

| Scheme | Acc (%) |
|---|---|
| Base (+CoT&KF) | 60.0 |
| +Best-of-N (Ours) | 63.3 |
| +Tree Search | 62.8 |

(c) Comparison of different sources of attention drift.

| Attn Source | Acc (%) |
|---|---|
| Base (+CoT) | 58.2 |
| + Video-Part | 59.9 |
| + Subtitles-Part | 60.0 |

Table 4: **Ablation Study on Sensor and Controller.** Accuracy (%) on VideoMMMU under different scoring policies and visual self-correction methods in Controller.

(a) Scoring policies. We set N=8 in Majority Voting and Score Forest.

| Scoring Method | Acc (%) |
|---|---|
| Base (+CoT) | 58.2 |
| Majority Voting | 61.9 |
| Score Forest (Ours) | 63.3 |

(b) Visual self-correction methods.

| Method | Acc (%) |
|---|---|
| Base (+CoT) | 58.2 |
| + Key Frames | 60.0 |
| + Dense Sampling | 60.3 |
| + Spatial Zoom-in | 60.7 |

Table 5: **Stability analysis under different disturbance levels.** "+Ours" refers to the N=2 setting.

| Setting | Base | +Ours |
|---|---|---|
| Uniform sampling | 55.0 | 60.0 |
| Disturb rate = 0.2 | 55.0 | 60.4 |
| Disturb rate = 0.4 | 55.4 | 60.2 |
| Disturb rate = 0.6 | 52.0 | 60.1 |

## 4.2 ABLATION STUDY ON MLLM INFERENCE SYSTEM AND SENSOR.

We conduct ablation studies on the VideoMMMU benchmark using Qwen2.5-VL-7B to investigate the contribution of each component in the CyberV framework.

**Performance boost via multiple inference strategies.** As shown in Table 3a, combining inference strategies like subtitles (+6.4%) and CoT prompting (+9.6%) significantly boosts performance. However, excessive reasoning may introduce distraction and attention drift. Our cybernetic loop alleviates this by incorporating attention-guided key frames, further improving accuracy to 60.0%. Note that here we adopt the **simplest form** of our framework: one base and one CoT response in the first round, followed by one response with key frames in the second round.

**BoN outperforms complex search schemes.** We justify our choice of Best-of-N (BoN), as it outperforms more complex strategies like PRM-guided tree search under a similar computational budget (Table 3b), confirming that BoN is a simpler yet more effective alternative for our framework.

**Sensor benefits from different attention sources.** The Sensor extracts intermediate signals like attention drift. As shown in Table 3c, incorporating subtitle-based attention drift alongside video-based drift offers a slight gain, indicating its complementary grounding value, though it may introduce noise when clear temporal anchors are absent.

## 4.3 ABLATION STUDY ON CONTROLLER.

**Score Forest outperforms majority voting.** We evaluate the Controller's ability to make decisions based on uncertainty signals. Under the BoN (N=8) setting, our score forest, which aggregates multi-dimensional uncertainty, outperforms simple majority voting, as shown in Table 4a. These results confirm the critical role of the Controller in our cybernetic loop.

**Different visual self-correction methods are effective.** Beyond direct key frame injection, we analyze other visual self-correction methods. As shown in Table 4b, temporal dense sampling and spatial zoom-in further improve performance, with zoom-in achieving the best result at 60.7%. These results validate the effectiveness of multi-dimensional visual scaling in improving model focus and answer accuracy. Due to the additional complexity of these methods, we use direct key frames injection in the main experiments. More ablation studies are provided in Appendix A.1.

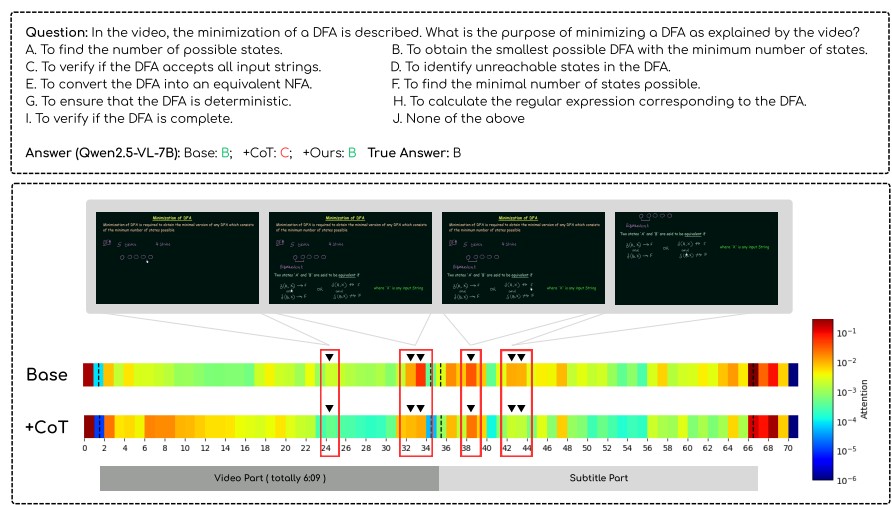

Figure 4: Attention map visualizations. Red boxes highlight segments where attention significantly drops after applying "CoT". They may correspond to content that contains critical information. Under our control system, adding key frames after "CoT" helps rectify previously incorrect responses.

### 4.4 STABILITY ANALYSIS

To assess the robustness of our framework, we conduct a stability analysis inspired by control theory. We introduce temporal perturbations by replacing uniform frame sampling with a non-uniform variant, where each frame index is randomly shifted within a range determined by the disturb rate. As shown in Table 5, our method consistently outperforms the baseline across all disturbance levels, even as the baseline degrades. These results demonstrate that our method is stable and robust to non-uniform temporal distortions, confirming the strength of our cybernetic test-time scaling strategy in dynamically adapting to sampling perturbations while maintaining reliable performance.

### 4.5 ANALYSIS OF PERFORMANCE VS. EFFICIENCY

To find an optimal balance between performance and cost of our framework, we analyzed the trade-off by varying the number of inference paths (N). As shown in Figure 3, accuracy peaks at 63.3% for N=8, while inference time grows sub-linearly due to fixed overheads like data preprocessing. Since increasing N beyond this point yields diminishing returns at a prohibitive computational cost, our analysis confirms that N=8 provides the best trade-off, which is adopted in our main experiments.

### 4.6 VISUALIZATION

Figure 4 illustrates the effectiveness of our control system in identifying forgotten yet critical visual information via attention difference after CoT. By reintegrating these cues, the system corrects CoT-induced errors, demonstrating the effectiveness of CyberV in boosting reasoning while preserving perception. More visualizations and limitations are provided in Appendix A.4 and A.5.

## 5 CONCLUSION

We propose CyberV, a training-free, extra-model-free, test-time adaptive scaling framework designed to enhance logical reasoning in video understanding with multimodal large language models (MLLMs). Inspired by cybernetic principles, CyberV integrates a closed-loop architecture with a MLLM Inference System-Controller-Actuator design to monitor attention shifts, evaluate prediction uncertainty, and dynamically execute self-correction strategies. Extensive experiments across diverse benchmarks demonstrate the effectiveness of CyberV, achieving substantial gains on tasks requiring both deep knowledge application and complex deductive reasoning. Future work will explore more effective and efficient strategies to further improve complex multimodal reasoning.

ETHICS STATEMENT

This work follows the ICLR Code of Ethics. All datasets used in our experiments, including VideoMMMU, MMVU, MMR-V, VideoMME, WorldSense and MVBench, are publicly available benchmarks. No private or sensitive information is involved, and the data usage strictly follows the intended academic licenses.

REPRODUCIBILITY STATEMENT

We provide detailed descriptions of the model architectures, inference strategies, and hyperparameter settings in the main paper and appendix. All datasets are public, and we will release our code to ensure full reproducibility.

LLM USAGE STATEMENT

Large language models (LLMs) are only used to polish the writing of this paper, including grammar correction, phrasing, and improving clarity of exposition. They do not contribute to research ideation, experimental design, data analysis, or the generation of results. All technical contributions and findings are entirely due to the authors.

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

# A  APPENDIX

**Overview.** In this appendix, we first provide more ablation studies on several component designs in A.1. Then, we discuss the results on perception-heavy benchmarks in A.2. Next, we include the PyTorch-style pseudo-code of our algorithm in A.3 to offer a clearer understanding of the overall inference process. After that, we present more visualization results in A.4. Finally, we discuss the limitations and future work in A.5.

## A.1  MORE ABLATIONS STUDIES

**Ablation on scoring mechanism in Score Forest.**

In our Score Forest implementation, the number of trees ($m$) is set to 5. The five scoring mechanisms are Attention Retention Score, Answer Confidence Score, Confidence Stability Score, Relative Rank Score and Text Repetition Score. Details of the five scores have been introduced in the main paper. According to Table 6, we conduct an ablation study on these five scores. Under a fixed random seed setting, the experiments demonstrate the effectiveness of each scoring mechanism, as removing any single one results in a drop in overall performance.

**Ablation study on attention extraction from different layers.** Recent studies in model interpretability suggest that the final layers of large language models tend to capture more high-level semantic information that directly contributes to the model's output decisions Ferrando et al. (2024). Motivated by this, we evaluate the effectiveness of extracting attention signals from different depths of the LLM component in Qwen2.5-VL-7B, which contains 28 transformer layers in total. Specifically, we experiment with extracting attention maps from the last 1, 4, and 7 layers, and report the results in Table 7. Extracting attention solely from the final layer yields an accuracy of 60.0%. Including the last 4 layers slightly reduces performance to 59.4%, while using the last 7 layers gives a marginal improvement to 60.2%. Overall, incorporating more layers introduces minor fluctuations, but yields consistent improvement over the baseline. For simplicity and computational efficiency, we finally extract attention only from the last layer in all experiments.

Table 6: Ablation study on the components of the Score Forest in our implementation. "w/o" stands for "without".

| Scoring Method | Accuracy (%) |
|---|---|
| Score Forest Baselines | **63.3** |
| w/o Attention Retention | 63.2 |
| w/o Answer Confidence | 62.7 |
| w/o Confidence Stability | 62.6 |
| w/o Relative Rank | 63.2 |
| w/o Text Repetition | 62.9 |

Table 7: Attention extraction from different layers. Here, we use Qwen2.5-VL-7B and compare the last 1st, 4th, and 7th layers.

| Attention from | Accuracy (%) |
|---|---|
| Base (No key frame) | 58.2 |
| Last 1 layer | 60.0 |
| Last 4 layers | 59.4 |
| Last 7 layers | 60.2 |

## A.2  PERFORMANCE ON PERCEPTION-HEAVY QUESTION-ANSWERING BENCHMARKS

Both strong perceptual abilities and logical deduction are essential for accomplishing complex video understanding tasks. However, while test-time scaling methods are intended to guide model reasoning, we find that **simple chain-of-thought (CoT) prompting may impair a model's direct perceptual capabilities**. We conduct tests on VideoMME Fu et al. (2025), WorldSense Hong et al. (2025), and MVBench Li et al. (2024c), three video understanding benchmarks where questions are predominantly focused on direct perception. As shown in Table 8, applying a simple CoT prompt significantly degraded the base model's performance across all three benchmarks, with drops of -2.3%, -2.1%, and -4.6%, respectively. This suggests that **perception and reasoning do not trivially enhance one another**; in fact, an unguided reasoning process can inhibit foundational perceptual accuracy.

Our CyberV framework, however, is designed to mitigate this issue through its adaptive selection and control mechanisms. The controller adaptively selects the most reliable response and triggers

Table 8: **Performance on perception-heavy video understanding benchmarks (accuracy %).** The base model is Qwen2.5-VL-7B. Subtitles are added in VideoMME and WorldSense. Unlike "CoT" strategy, **CyberV maintains the base model's perception performance.**

| Model | VideoMME | WorldSense | MVBench |
|---|---|---|---|
| Base | 70.5 | 46.0 | 66.7 |
| +CoT | 68.2 (-2.3) | 43.9 (-2.1) | 62.1 (-4.6) |
| +Ours | **71.6 (+1.1)** | **47.1 (+1.1)** | **67.5 (+0.8)** |

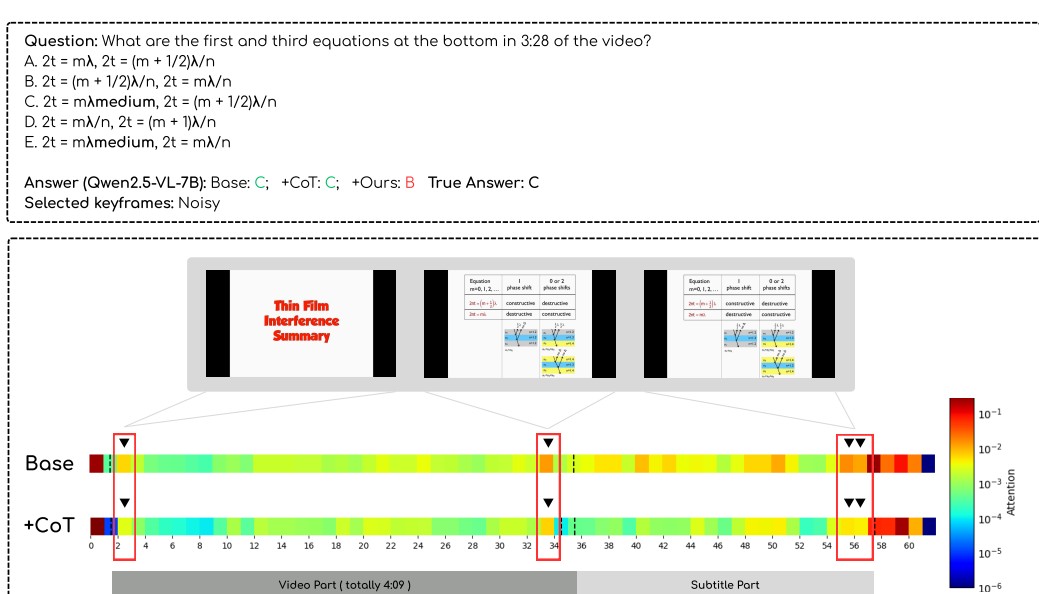

Figure 5: Without confidence-based filtering (Score Forest) in the Controller, high-confidence correct answers in the first round also need to trigger unnecessary key frames extraction, leading to errors in the second round due to noisy frames. In this case, although the question refers to 3:28 of a 4:09 video, the selected key frames focus on the beginning and end, resulting in an incorrect revision.

self-correction when necessary. As a result, unlike the fixed CoT approach, **our method avoids weakening the model's perceptual abilities** and even yields slight performance gains of +1.1% on VideoMME, +1.1% on WorldSense, and +0.8% on MVBench. These outcomes underscore the necessity and utility of incorporating a self-correction mechanism. Along with results in the main paper, these findings demonstrate that CyberV can enhance a model's logical reasoning capabilities while preserving its foundational perceptual skills, thereby enabling more complex video question-answering tasks.

## A.3 PSEUDO-CODE OF CYBERV

Figure 8 illustrates the PyTorch-style pseudo-code of the CyberV architecture, which models video reasoning as a closed-loop control process. It consists of three core modules: the MLLM Inference System executes various scaling strategies; the Sensor monitors intermediate outputs and extracts key signals such as predictions and attention drift; and the Controller evaluates response reliability and, if needed, constructs feedback (e.g., key frames augmentation) to trigger self-correction. These components interact iteratively to improve reasoning quality without additional training.

## A.4 MORE VISUALIZATION RESULTS

**More attention map visualizations.** In the main paper, we show that incorporating attention-guided key frames into the second-round inference can effectively correct CoT-induced errors, particularly in

cases where the base model initially produces the correct answer but CoT leads to an incorrect one. This demonstrates the utility of the cybernetic feedback loop in mitigating reasoning drift.

However, as illustrated in Figure 5, applying visual self-correction indiscriminately (without confidence-based filtering) can introduce new errors. In this example, both the base model and CoT initially provide the correct answer with high confidence, yet the second round reversed the decision due to the influence of noisy and irrelevant key frames. The contrast between this failure case and successful examples underscores the importance of incorporating confidence-aware control to selectively trigger feedback only when necessary, thereby enhancing overall robustness and preserving performance on easy cases.

**Case studies on MMR-V.** We conduct visualization studies on the MMR-V benchmark to qualitatively evaluate the effectiveness of our model. As depicted in Figures 6 and 7, these case studies illustrate both the strengths and limitations of the proposed cybernetic loop.

In Figure 6, a scenario is presented where both the base model and the Chain-of-Thought (CoT) approach fail, but our proposed method succeeds in comprehending the nuanced context of a video. Our method effectively leverages selected key frames that encapsulate the critical events, enabling it to deduce the correct, more complex situation: visiting a seriously ill boyfriend to wish him well. This case highlights our method's strength in identifying and focusing on the most relevant temporal segments of a video to overcome misleading information from "CoT" and achieve accurate reasoning.

Figure 7 demonstrates a failure case where the performance of our model is degraded by noisy or irrelevant frames. This case shows a limitation of our method: inaccurate key frames extraction may, in certain instances, hinder the effectiveness of second-round visual enhancement, thereby failing to support meaningful self-correction.

## A.5 LIMITATIONS AND FUTURE WORK DISCUSSION

While CyberV demonstrates notable improvements in test-time video reasoning, several limitations remain. First, the current key frames extraction relies on attention drift over video and subtitle segments. Although some critical frames are often covered, this approach may **introduce noisy or irrelevant frames**. Our cybernetic loop can mitigate their impact through selective correction, but more principled methods for noise filtering, temporal search and the utilization of signals remain important directions for future work. Second, **current state-of-the-art MLLMs exhibit limited capacity for temporally grounded perception and reasoning**. That is, the ability to precisely align and integrate the information from visual frames, subtitles, and questions along the temporal axis during the reasoning process. We believe that combining CyberV with future MLLMs possessing stronger multi-modal spatio-temporal understanding capabilities may yield greater benefits. Another limitation lies in **inference efficiency**. As the number of inference paths ($N$) and iterations increases, test-time latency grows manyfold. Developing more efficient implementations of the cybernetic loop, potentially via strategy pruning, presents another valuable avenue for future research.

Overall, while CyberV opens up a novel perspective on test-time adaptive reasoning, its full potential can be further unlocked through improvements in both base model capability and control system efficiency.

| Good Case: | CyberV boosts model preformance |
|---|---|
| Benchmark: | MMR-V |

**Question:** What might the girl be writing a greeting card for?
  **A**. Valentine's Day celebration.          **B**. Milo's birthday.
  **C**. A graduation celebration.              **D**. Apology for a mistake.
  **E**. Milo's wedding anniversary.           **F**. A heartfelt reunion with Milo.
  **G**. A thank you for help with a project.   **H**. Christmas holiday greeting.
  **I**. A shared achievement celebration.
  **J**. To visit her seriously ill boyfriend and wish him well.
  **K**. A farewell for a move to a new city.

**Answer: J.**

**Qwen2.5-VL-7B:** A.

**Qwen2.5-VL-7B (+CoT):**

*Thinking process:*
1. The setting includes a bouquet of flowers, which often symbolize affection or care.
2. The girl is writing on a card, implying a personal message is being composed.
3. There is an envelope, suggesting this could be a message meant to be sent.
4. The mention of "for Milo" indicates the recipient of the message.
*Final answer:* F.
*Remark:* The key events were not accurately identified, leading the model to focus on misleading information such as "for Milo".

**Qwen2.5-VL-7B (Ours):**

*Best-of-8 output：* 【A, B, F, J, B, B, F, F】
Uncertain!
*Selected keyframes:*

*Final answer:* J.
*Remark:* The selected keyframes cover the timestamps when the events mentioned in the question occur.

Figure 6: Base model and "CoT" fail to answer but our method performs well.

| Bad Case: | Noisy frames may degrade model preformance |
|---|---|
| Benchmark: | MMR-V |

**Question:** What message does the entire video intend to convey?
**A.** Having advanced abilities makes someone superior in all situations.
**B.** Playing soccer requires extraordinary physical talent.
**C.** People should fear what they don't understand.
**D.** Strength always wins respect from others.
**E.** Inclusion and acceptance can transcend differences, even across major divides like human and non-human.
**F.** Team sports are the only way to bring people together.
**G.** Distinguishing oneself through physical displays should be everyone's goal.
**H.** This telecommunications company has a strong signal transmission capability.
**I.** Childhood friendship is only about sports.
**J.** A mysterious figure is always a potential danger.
**K.** Robots are better at physical activities than humans.

**Answer: H.**

**Qwen2.5-VL-7B: D.**

**Qwen2.5-VL-7B (+CoT):**

*Thinking process:*
1. The video shows a group of children playing soccer, with one child performing an acrobatic kick that causes another child to fall off a balcony.
2. The video ends with a logo for Ooredoo and a message promoting their brand.
*Final answer:* **H.**

**Qwen2.5-VL-7B (Ours):**

*Best-of-8 output：* 【D, D, H, E, F, E, F, E】
Uncertain!
*Selected keyframes:*

*Final answer:* **F.**
*Remark:* The selection of keyframes is not accurate, which may lead to incorrect final answer.

Figure 7: Noisy frames may degrade model performance.

**Pseudo-code of the CyberV architecture.**

```python
# Core module: Sensor
# signal extraction from MLLM forward
    processes
class Sensor:
    def monitor(outputs):
        signals = []
        signals.append(get_predicitons(
    outputs))
        signals.append(get_attn_drift(
    outputs))
        ...
        return signals

# Core module: Controller
# decision making and feedback construction
class Controller:
    def score_forest(out, attn_drift):
        # cal score for each response
        # aggregate scores for each option
        return score_list

    def inference_feedback(signals):
        # key frames extraction
        video_based_kfs, sub_based_kfs = ...
        kf =  video_based_kfs |
    sub_based_kfs
        # visual cues enhancement
        # Here is an example: directly
    augment key frames
        action = ("Add key frames.", kf)
        return action

    def decide(signals):
        scores = score_forest(signals)
        response = best_answer(scores)
        if is_confident(scores):
            return True, response, None
        else:
            action = inference_feedback(
    signals)
            return False, response, action
```

```python
# Core module: MLLM Inference System
# Executing Inference Strategies
class MLLMSystem:
    # an MLLM. For example, Qwen2.5-VL-7B
    self.model = ...
    def execute(inputs):
        outputs = []
        # Use BoN to get N responses
        for s in inputs['strategies']:
            out = self.model.forward(inputs,
     s)
            outputs.append(out)
        return outputs

# Cybernetic loop for one inference round
def run_one_loop(inputs):
    # Step 1: Run MLLM with multiple
     strategies
    out = MLLMSystem.execute(inputs)
    # Step 2: Monitor outputs to extract
     signals
    signals = Sensor.monitor(out)
    # Step 3: Decide when & how to trigger
     self-correction
    flag, response, action = Controller.
     decide(signals)
    return flag, response, action

# Closed-loop inference over multiple rounds
def cyber_v(inputs):
    max_rounds = ...
    round_now = 1
    while True:
        flag, response, action =
     run_one_loop(inputs)
        if flag or round_now == max_rounds:
            return response
        inputs = update(inputs, action)
        round_now += 1
```

Figure 8: Pseudo-code of the CyberV architecture. The MLLM Inference System, Sensor and Controller cooperate to form a closed-loop control cycle for test-time video understanding.

