# OpenReview forum: "CyberV: A Cybernetic Framework for Enhancing Logical Reasoning in Video Understanding"
_ICLR.cc/2026/Conference — ICLR 2026 Conference Withdrawn Submission_

### Official Review · Reviewer_2how · 2025-10-15

**Soundness:** 3
**Presentation:** 2
**Contribution:** 2
**Rating:** 4
**Confidence:** 3

**Summary:**

This paper addresses the limitations of current Multimodal Large Language Models (MLLMs) in deep logical reasoning for video understanding—such as feed-forward processing constraints (lack of self-correction), poor test-time scaling, and hallucinations. Inspired by cybernetic principles (control, communication, self-regulation), it proposes CyberV, a training-free, test-time adaptive scaling framework that redesigns video MLLMs into closed-loop adaptive systems.

**Strengths:**

Training-Free & Model-Agnostic: Enhances frozen MLLMs without retraining, fine-tuning, or adding auxiliary models—reducing computational costs and enabling easy integration with existing video MLLMs (e.g., Qwen2.5-VL, InternVL3).

Robust Logical Reasoning: Addresses MLLMs’ feed-forward limitations via closed-loop self-monitoring and self-correction, outperforming proprietary models (GPT-4o) even with small open-source models (7B parameters).

Preserves Perceptual Ability: Unlike naive CoT prompting (which degrades perception by 2.1–4.6% on perception-heavy benchmarks like MVBench), CyberV’s adaptive control maintains or slightly improves perceptual accuracy (+0.8–1.1%).

**Weaknesses:**

1. Key Frame Extraction Noise: Relies on attention drift to select key frames, which may include irrelevant or noisy frames (e.g., misaligned timestamps in subtitles). This can lead to incorrect self-correction in edge cases (e.g., reversing correct answers due to noisy inputs).
2. Limited Temporal Grounding: Relies on base MLLMs’ ability to align visual frames, subtitles, and questions temporally. If the base model lacks strong spatio-temporal understanding, CyberV’s feedback may fail to target critical temporal segments.
3. Inference Efficiency Trade-Off: While N=8 is optimal, increasing inference paths (N) or rounds leads to significant latency growth. More detailed computational cost should be discussed.
4. Rule-Based Scoring Limitation: The Score Forest uses hand-crafted rules (e.g., fixed text repetition thresholds) rather than learned metrics. This may limit adaptability to diverse video tasks (e.g., free-form questions, not just MCQs).
5. The method seems general, why not apply it to llm but only focus on mllm?

**Questions:**

please refer to the weakness part.

---

### Official Review · Reviewer_pTZ8 · 2025-10-26

**Soundness:** 2
**Presentation:** 3
**Contribution:** 2
**Rating:** 4
**Confidence:** 4

**Summary:**

CyberV proposes a test-time, control-theoretic framework to boost logical reasoning in video understanding without any additional training. It runs a Best-of-N (BoN) set of reasoning paths (base + multiple CoT variants), uses a “Sensor” to measure attention drift between base and CoT answers (from the last-layer attention of the answer token to video/subtitle segments), and a “Controller” (Score Forest) to aggregate multi-signals (attention retention, confidence, stability, rank, repetition) into a TopScore that decides whether to stop or trigger feedback. When uncertain, CyberV performs targeted inference feedback by extracting key frames from segments with the largest negative drift (optionally with dense temporal sampling or spatial zoom-in) and re-injects them for a second round (N=1) to correct evidence usage. Across VideoMMMU, MMVU-MCQ, and MMR-V, the method consistently improves accuracy—often substantially for small open-source MLLMs—and avoids the perception degradation that naïve CoT can cause on perception-centric benchmarks. The approach emphasizes a lightweight, training-free, closed-loop that couples evidence perception with reasoning, showing strong performance-efficiency trade-offs (e.g., peak gains around N=8) and pointing to future work on more robust feedback selection and broader free-form generation.

**Strengths:**

1. This paper proposes a simple, training-free, test-time closed-loop that marries sensing (attention-drift), control (Score-Forest gate), and feedback (key-frame injection), giving a clear, modular recipe that others can reproduce.

2. This paper grounds “test-time scaling” in control theory rather than ad-hoc prompt tricks, articulating why feedback should be triggered conditionally by model uncertainty and evidence drift.

**Weaknesses:**

1. Lack essential experimental setting details. This paper tests a bunch of baselines in Table 1. However, paper didn't specify how these baseline results are collected, eg, temperature, token length, number of samples, w/wo reasoning, etc.

2. This paper’s key-frame selection can inject noisy/irrelevant frames; without careful gating it may even flip a correct first-round answer to wrong in round two.

**Questions:**

1. Why is CyberV set for 2 iterations? What will happen if more iteration is introduced?

2. How is the hyper-parameter N=8 and $\tau$=0.3 chosen, and what will be the influence for other number combinations?

3. Can you explain Figure 3 more?

4. It would be helpful to add the specific model name for base on Table 3, 4, for easier understanding.

---

### Official Review · Reviewer_DTdm · 2025-10-30

**Soundness:** 2
**Presentation:** 3
**Contribution:** 2
**Rating:** 4
**Confidence:** 5

**Summary:**

This paper designed a test-time scaling framework inspired by cybernetics, consisting of a MLLM, a sensor and a controller which are working together to determin the execution path of MLLM in multimodal reasoning. Experiments suggests that this framework can significantly improves the accuracy of esxisting MLLMs on certain benchmarks.

**Strengths:**

● The idea of intrucuding cybernetics into MLLM systems to control their test-time behavior is interesting.
● The design of MLLM-Sensor-Controller makes sense.

**Weaknesses:**

● The proposed framework inutuationly increases the inference cost significantly even compared to other test-time scaling methods, as it incoporates multiple inferencing for MLLM w. and w.o long CoT, with possiple re-execution loop.
● The authors claims that the attention drift between base and cot model indicates the degraded perceptual grounding, but this key insight is not verified by quantitive or quailitative results. Moreover, why is the attention score of the final layer is adopted rather than other layers? The motivation of selecting only the last layer is insufficent.
● The description of the proposed method is not clear. In example, in Sec. 3.2, the authors claimed sensors also collect other signals beyond attention drift, but no details are provided. Moreover, how are the attention scores calculated is also not described: are they averaged on all output tokens or only calculated on some specific tokens? or In Sec. 3.3, only names of the scores are listed, and how to calculate them is ignored except the Relative Rank Score. Such insufficent description on the key contribution of the paper significantly blocks the readers from reproducing the results and undermines the soundness of this paper.
● The evaluation benchmark is limited. The experiment only comprehensively verifies the proposed framework on VideoMMMU and gives sketched results on MMVU and MMR-V. Beyond the insufficent baselines on MMVU and MMR-V, more benchmarks are necessary to access the general effectiveness of the proposed baselines. The performance on more tasks, such as TempCompass, TOMATO and EgoSchema, and more general scenes, such as MVBench and LMVU, should be verified.

**Questions:**

● What is the cost of proposed method? How is it cost and improvement compared to existing test-time scaling methods?
● Are there qualitative and quantitave verifications on the claim that attention drift indicates the degraded perceptual grounding?
● What is the motivation of using only last layer to calculate the attention drift of visual content input tokens?
● How is the attention calculated in attention drift? How are the other signals that sensor colloected calculated other than the attention drift? How are the scores calculated in controller?
● What is the performance of the framework on more general scene and more tasks? Does the framework still boost the performance on various model on MMVU and MMR-V?

---

### Official Review · Reviewer_V9FT · 2025-11-01

**Soundness:** 3
**Presentation:** 3
**Contribution:** 1
**Rating:** 4
**Confidence:** 4

**Summary:**

This paper introduces **CyberV**, an approach that leverages cybernetic structures to enhance the reasoning performance of Multi-Modal Large Language Models (MLLMs).

**Strengths:**

- The authors successfully extend cybernetic theory to the MLLM domain, designing a **closed-loop feedback mechanism** that dynamically incorporates key frames into the reasoning process based on response scores.
- The overall framework is interesting, offering a **training-free solution** that is straightforward and easy to understand.
- Experimental results demonstrate that **CyberV significantly enhances the reasoning capabilities** of models across three relevant benchmarks.

**Weaknesses:**

### 1. Attention Mechanism and Segment-Level Scores

- In Section 3.3, the authors define the attention scores between the answer token and the video/subtitle segments.
- Traditionally, attention scores are computed **between tokens**, not between a token and a segment.
- The paper describes the attention scores as:
$$
  A_{\text{video}}^h \in \mathbb{R}^{1 \times K_1}, \quad A_{\text{sub}}^h \in \mathbb{R}^{1 \times K_2}
$$
  where $K_1$ and $K_2 $represent the number of video and subtitle segments, respectively.
- **Clarification requested**: How are these segment-level attention scores computed? Are they obtained by **averaging or summing token-level attention scores** across each segment? A detailed explanation would be appreciated.

### 2. Assumptions in Equation (1)

- Equation (1) suggests that the base model's attention reflects a more direct, foundational grounding on visual evidence.
- **Concern**: This assumption may not always hold. The base model might not always correctly identify the most relevant segments or tokens during inference.
- It seems Equation (1) may primarily highlight differences in attention between the CoT and base models, rather than supporting the claim that the base model's attention is "distracted."
- **Request**: Clarification and case studies/examples to substantiate this claim.

### 3. Use of Answer Token in CoT Mode

- In CoT mode, attention scores are calculated using the **answer token**.
- **Concern**: The answer token represents the final output, not the intermediate reasoning steps.
- It may be more appropriate to compute attention based on **intermediate tokens** that represent reasoning at various stages.
- **Request**: Clarify why the answer token is used instead of intermediate tokens.

### 4. Computational Cost and Resource Consumption

- Experiments were conducted using **eight 80GB GPUs**.
- Although CyberV is training-free, it still incurs **significant computational cost**.
- **Request**: Provide information on average inference time, scalability, and feasibility for **real-time or resource-constrained environments**.

### 5. Re-Embedding Key Frames and Reasoning Process

- The paper describes **re-embedding key frames** into the input.
- **Concern**: It's unclear whether re-embedding directly produces the output or if it is guided by an underlying **CoT structure**.
- **Request**: Clarify whether re-embedding directly leads to the answer or is part of CoT reasoning. Providing a **sample prompt template** would be helpful.

### 6. Effect of Segment Division on Performance

- The segmentation mechanism divides video and subtitle inputs into smaller segments.
- **Concern**: The impact of the number of segments on reasoning performance is not addressed.

**Questions:**

- If \(K\) is too small, could information be **redundant or insufficient**?
- If \(K\) is too large, could the model **fail to capture critical segments**?
- **Request**: Provide an experimental analysis of how segment division affects reasoning performance and discuss **optimal parameter settings**.

---

### Note · Authors · 2025-11-26

I have read and agree with the venue's withdrawal policy on behalf of myself and my co-authors.